# Comparative Analysis of the Complete Chloroplast Genomes of Six Endangered *Cycas* Species: Genomic Features, Comparative Analysis, and Phylogenetic Implications

Jianmin Tang [1,†], Rong Zou [1,†], Taiguo Chen [1,2], Lipo Pan [1,3], Shujing Zhu [1,3], Tao Ding [1], Shengfeng Chai [1] and Xiao Wei [1,*]

1   Guangxi Key Laboratory of Plant Functional Substances and Resources Sustainable Utilization, Guangxi Institute of Botany, Guilin 541006, China; tjm@gxib.cn (J.T.); zr@gxib.cn (R.Z.); 19178304083@163.com (T.C.); panlipo2021@163.com (L.P.); 17377087031@163.com (S.Z.); dt@gxib.cn (T.D.); csf@gxib.cn (S.C.)
2   School of Pharmacy, Guilin Medical University, Guilin 541000, China
3   College of Life Sciences, Guangxi Normal University, Guilin 541006, China
*   Correspondence: weixiao@gxib.cn
†   These authors contributed equally to this work.

**Abstract:** *Cycas* (family Cycadaceae), which spread throughout tropical and subtropical regions, is crucial in conservation biology. Due to subtle morphological variations between species, a solid species-level phylogeny for *Cycas* is lacking. In the present study, we assembled and analyzed the chloroplast genomes of six *Cycas* plants, including their genome structure, GC content, and nucleotide diversity. The *Cycas* chloroplast genome spans from 162,038 to 162,159 bp and contains 131 genes, including 86 protein-coding genes, 37 transfer RNA (tRNA) genes, and 8 ribosomal RNA (rRNA) genes. Through a comparative analysis, we found that the chloroplast genome of *Cycas* was highly conserved, as indicated by the contraction and expansion of the inverted repeat (IR) regions and sequence polymorphisms. In addition, several non-coding sites (psbK-psbI, petN-psbM, trnE-UUC-psbD, ndhC-trnM-CAU, and rpl32-trnP-GGG) showed significant variation. The utilization of phylogenetic analysis relying on protein-coding genes has substantiated the division of *Cycas* primarily into four groups. The application of these findings will prove valuable in evaluating genetic diversity and the phylogenetic connections among closely related species. Moreover, it will provide essential support for the advancement of wild germplasm resources.

**Keywords:** *Cycas*; chloroplast genome; phylogenetic analysis; SSR; large repeats

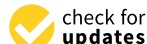



## 1. Introduction

Cycads are one of the oldest families of living seed plants, dating back to the Late Permian period [1]. More than 360 species of cycads exist, which have been divided into 10 genera and 2 families, Cycadaceae and Zemitaceae [2]. These plants are found in patches throughout the tropical and subtropical regions of Asia, Africa, Oceania, and America [3]. From a morphological perspective, the reproductive systems of cycads are most similar to those of spore plants, making cycads very important in studying the genesis and early evolution of seed plants. From the standpoint of their phylogenetic origin and evolution, cycads have existed and reproduced for at least 280 million years [4], suffered significant environmental changes, and contain a wealth of genetic data. In terms of species protection, cycads have persisted and flourished until the present, although many plant species have perished due to changes in the earth's environment. Therefore, studying the mechanisms underlying the environmental adaptability of cycads is critical in conservation biology.

The genus *Cycas* is the sole representative within the Cycadaceae family, encompassing approximately 120 species and exhibiting a complex taxonomy across a diverse geographical range [4]. *Cycas* are thought to have originated in Tertiary East Asia according to recent

biogeographic research [5]. In China, there are roughly 20 different species of *Cycas*, most of which are found in the southwest and southeast coasts [6].

There have been some challenges and disagreements in the categorization of the *Cycas* species for the following reasons. First, it is challenging to properly analyze the physical traits of *Cycas*. Since these species take a long time to reach sexual maturity, there are few flowering plants to be discovered. When combined with variable environments, populations of the same *Cycas* species can also differ significantly. Additionally, natural hybridization may exist between different species [7]. Hill proposed dividing *Cycas* into six groups based on the long-term observations of cycad plants in the field and integrating reproductive characteristics such as the ovule coat, megaspore leaf shape, and anatomical structure of the seeds: Section Asiorientales, Section Strangeriodes, Section Indosinenses, Section Cycas, Section Panzhihuaenses, and Section Wadeae [8,9]. Recently, Zheng et al. proposed 23 species from four sections in China based on distribution and morphological characteristics [6]. Traditional taxonomy, however, has a limited ability to establish species boundaries due to the small morphological variations between species induced by either environmental or genetic causes.

The strategy of integrating DNA identification and morphological features was also employed to define *Cycas* species, and more accurate findings were achieved. Using molecular sequencing techniques, Xiao and Möller performed a phylogenetic study of 31 *Cycas* species using the nrDNA ITS gene [10]. The chloroplast (cp) genome presents substantial advantages for investigations in the field of plant evolutionary biology due to its genetic stability, well-preserved genome structure, and rate of evolutionary change that outpaces that of mitochondria [11]. Liu et al. used four chloroplast genes and three nuclear genes to conduct a phylogenetic analysis of 104 species and 5 subspecies in the genus *Cycas* [12]. Nonetheless, the exploration of genomic resources within this genus has remained relatively limited, as evidenced by the small number of studies conducted [13–15]. In the GenBank database, the collection of complete cp genome sequences for *Cycas* species is currently limited to approximately 10 entries.

In this study, the cp genomes of six *Cycas* species, *C. longlingensis*, *C. longisporophylla*, *C. guizhouensis*, *C. ferruginea*, *C. crassipes*, and *C. bifida*, were sequenced. The objectives of this study are to (I) characterize the structures of the six newly sequenced chloroplast genomes of *Cycas*, (II) conduct a comparative analysis of the cp genomes among Cycas species, and (III) reconstruct the phylogenetic relationships within the Cycadaceae family using plastome sequences for selected species. Our results complement current genetic information on *Cycas* species and serve as a good reference for *Cycas* DNA molecular research.

## 2. Materials and Methods

### 2.1. Plant Materials and DNA Extraction

The fresh leaves of six different *Cycas* species, *C. longlingensis*, *C. longisporophylla*, *C. guizhouensis*, *C. ferruginea*, *C. crassipes*, and *C. bifida*, were taken from the Guilin Botanical Garden (Guangxi, China; coordinates: N 25°4′14.88″, E 110°17′57″) and immediately placed in liquid nitrogen. The total genomic DNA was extracted from fresh leaves (>1.0 g) using a Magnetic Plant Genomic DNA Kit (TIANGEN Biotech, Beijing, China) according to the manufacturer's instructions. The quality of DNA was evaluated using a TBS-380 Mini-Fluorometer (Invitrogen, MA, USA) and electrophoresis on a 1% agarose gel.

### 2.2. Chloroplast Genome Sequencing and Assembling

A total of 1 µg DNA was utilized as an input material for library construction. Using the VAHTS Universal Plus DNA Library Prep Kit for Illumina (Vazyme, Nanjing, China), we crafted sequencing libraries in accordance with the manufacturer's guidelines while applying index codes to individual sample sequences. Briefly, the DNA sample underwent initial fragmentation into 300–500 bp segments through sonication. Subsequently, the pre-existing DNA fragments underwent end-polishing and A-tailing, followed by ligation

with full-length adaptors for sequencing. Polymerase chain reaction (PCR) amplification was then performed using a cBot Truseq PE Cluster Kit v3-cBot-HS (Illumina). Lastly, PCR products underwent purification using an AMPure XP system (Beckman Coulter Inc., Brea, CA, USA); their library size distribution was determined using an Agilent 2100 Bioanalyzer, and quantification was carried out via real-time PCR. Using a cBot Cluster Generation System (Illumina Inc.), the indexed samples underwent clustering according to the manufacturer's protocols. Following clustering, the resulting libraries underwent sequencing on an Illumina Novaseq 6000 platform, generating reads with a length of 150 bp in the paired-end configuration.

The raw paired-end reads were subjected to quality assessment using FastQC v0.11.7 software. After quality evaluation, the obtained data were processed through de novo assemblers (Fast-plast, https://github.com/mrmckain/Fast-Plast, accessed on 8 May 2023, or GetOrganelle, https://github.com/Kinggerm/GetOrganelle, accessed on 8 May 2023) to generate optimal contigs. The cp sequence of *C. szechuanensis* (MH341576) was retrieved from GenBank and employed as the reference seed sequence for *C. longlingensis*, *C. longisporophylla*, *C. ferruginea*, *C. crassipes*, and *C. bifida*. Additionally, the cp sequence of C. bifida (MW900434) was utilized as the seed sequence for *C. guizhouensis*.

Subsequently, the chloroplast (cp) genomes underwent annotation using the PGA (available at https://github.com/quxiaojian/PGA, accessed on 10 May 2023) and the Geseq (https://chlorobox.mpimp-golm.mpg.de/geseq.html, accessed on 10 May 2023) with default settings, followed by manual corrections. The resulting gene maps were visualized using the online tool OGDraw v1.2 [16]. Six newly sequenced complete cp genomes were deposited to GenBank with the following Accession Numbers: Accession Nos. *C. bifida* (OQ862764), *C. crassipes* (OQ862765), *C. ferruginea* (OQ862766), *C. guizhouensis* (OQ862767), *C. longisporophylla* (OQ862768), and *C. longlingensis* (OQ862769).

### 2.3. Repeat Sequences and SSRs

The cp genome sequences of *C. szechuanensis* (NC_064393.1), *C. shiwandashanica* (NC_064393.1), and *C. segmentifida* (NC_064393.1) downloaded from GenBank were coupled with six newly sequenced cp genomes from *Cycas* to conduct an analysis of repeat sequences and simple sequence repeats (SSRs). We utilized a Perl script named MISA to identify SSRs within the complete cp genome sequences of the nine *Cycas* species. Specific threshold values were set for different SSR lengths, including mononucleotides, dinucleotides, trinucleotides, tetranucleotides, pentanucleotides, and hexanucleotides, with thresholds established at 10, 6, 5, 5, 5, and 5, respectively. Furthermore, the REPuter program was used to identify four categories of repeat sequences: palindromic, forward, reverse, and complement repeats. The recognition of repeat sequences adhered to the following criteria: (1) A Hamming distance of 3; (2) a minimum size of 20 bp; (3) a sequence identity equal to or greater than 90%.

### 2.4. Comparison of Whole Chloroplast Genomes and Divergent Hotspot Identification

The mVISTA comparative genomics server was utilized to generate a sequence variation map with an annotation of the *C. bifida* cp genome as a reference. An evaluation of IR sequence variations, encompassing features such as expansion and contraction, was conducted through the IRscope online program (https://irscope.shinyapps.io/irapp/, accessed on 8 May 2023). To pinpoint regions of intergeneric divergence, we conducted a sliding window analysis using DnaSP v5.10 software [17]. This analysis involved a window length of 600 bp with a step size of 200 bp.

### 2.5. Phylogenomic Reconstruction

We conducted a phylogenomic analysis utilizing six newly sequenced cp genomes of the *Cycas* species. Additionally, we included eight *Cycas* species sourced from GenBank in our analysis. These sequences were used to build a phylogenetic tree with *Encephalartos lehmannii* and *Bowenia serrulata* as outgroups, using the maximum-likelihood (ML) ap-

proach. To reconstruct ML trees, we extracted 86 protein-coding genes from the 16 species. MAFFT was used for multiple sequence alignment [18], and the GTR-GAMMA (GTR + G) model [19] was chosen based on a model test using the Bayesian information criterion (BIC) [20]. The MEGA-X v10.2.6 software facilitated the execution of maximum likelihood (ML) trees, with 1000 bootstrap replicates being configured to assess the branch support values. FigTree v1.4.4 was used to visualize the generated phylogenetic tree.

## 3. Results

### 3.1. Genome Structure of Chloroplast

The chloroplast genome of *Cycas*, ranging in size from approximately 162,038 to 162,159 bp, exhibits a general structural similarity to chloroplast genomes of other gymnosperm plants. It follows a quadripartite organization, comprising a large single-copy (LSC) region, a small single-copy (SSC) region, and two inverted repeat (IR) regions (Figure 1). The length of the LSC region ranged from 88,819 to 88,946 bp (Table 1). The GC content in the LSC regions remained fairly consistent across six species, varying from 38.70% in C. bifida to 38.73% in *C. longisporophylla*. The length of the SSC region fell within the range of between 23,102 and 23,124 bp, with a GC content of 36.52%–36.55%. The length range for the IR regions of the six *Cycas* species was 25,049–25,060 bp, which contained 42.01%–42.03% GC content. The total number of genes was 131, including 86 protein-coding genes, 37 transfer RNA (tRNAs), and 8 ribosomal RNA (rRNAs) in the cp genomes of all six *Cycas* species. The overall organization and structure of the *Cycas* chloroplast genomes were similar to those of other seed plant chloroplast genomes, with a conserved set of genes involved in photosynthesis and other chloroplast functions.

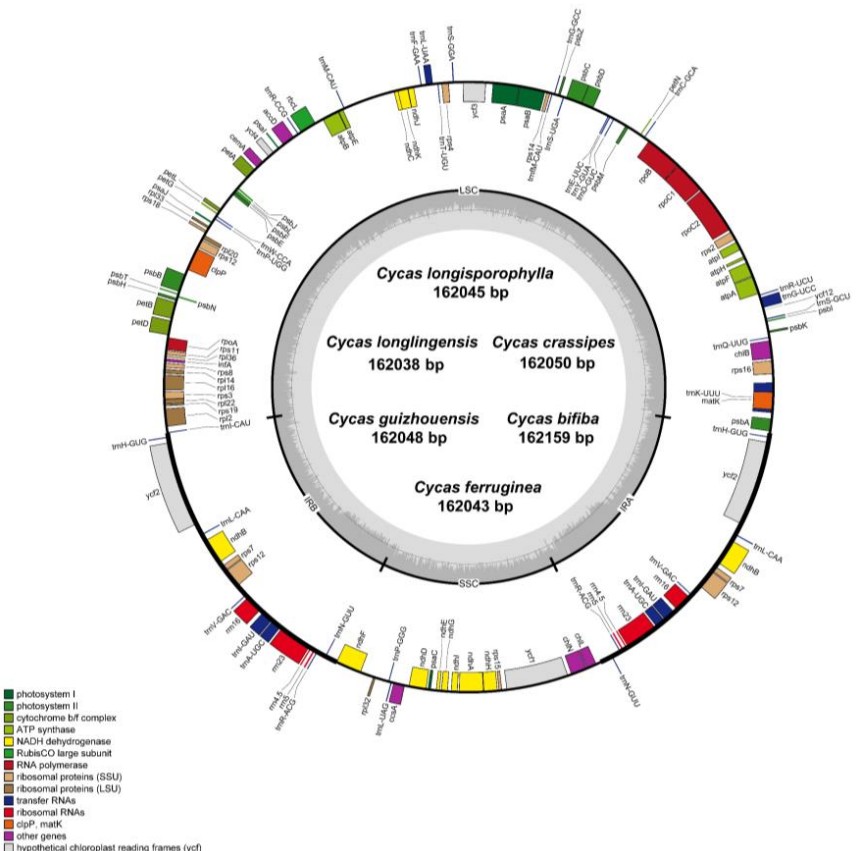

**Figure 1.** Gene map of the chloroplast genomes of six *Cycas* species. Genes located on the outer circle are transcribed in a clockwise direction, while those on the inner circle are transcribed counter-clockwise. Genes categorized into distinct functional groups are color-coded for easy identification. The inner circle exhibits variations in shading, with darker gray representing the GC content of the chloroplast genome and lighter gray corresponding to the AT content.

**Table 1.** Statistics on the basic features of chloroplast genomes in six *Cycas* species.

| Sample | Total Genome | | LSC | | IR | | SSC | | Gene Number | | | |
|---|---|---|---|---|---|---|---|---|---|---|---|---|
| | Length (bp) | G+C Content (%) | Length (bp) | G+C Content (%) | Length (bp) | G+C Content (%) | Length (bp) | G+C Content (%) | Total | PCGs | tRNA | rRNA |
| *Cycas longisporophylla* | 162,045 | 39.44 | 88,823 | 38.73 | 25,049 | 42.03 | 23,124 | 36.55 | 131 | 86 | 37 | 8 |
| *Cycas bifida* | 162,159 | 39.42 | 88,946 | 38.7 | 25,053 | 42.02 | 23,107 | 36.52 | 131 | 86 | 37 | 8 |
| *Cycas guizhouensis* | 162,048 | 39.42 | 88,826 | 38.71 | 25,060 | 42.01 | 23,102 | 36.54 | 131 | 86 | 37 | 8 |
| *Cycas crassipes* | 162,050 | 39.42 | 88,828 | 38.71 | 25,060 | 42.01 | 23,102 | 36.54 | 131 | 86 | 37 | 8 |
| *Cycas longlingensis* | 162,038 | 39.44 | 88,819 | 38.72 | 25,049 | 42.03 | 23,121 | 36.55 | 131 | 86 | 37 | 8 |
| *Cycas ferruginea* | 162,043 | 39.43 | 88,823 | 38.72 | 25,049 | 42.03 | 23,122 | 36.55 | 131 | 86 | 37 | 8 |

### 3.2. IR Expansion and Contraction

The IR border regions were compared according to the chloroplast genome sequences and annotation data for the six newly sequenced *Cycas* species, along with three other *Cycas* species: *C. szechuanensis*, *C. shiwandashanica*, and *C. segmentifida* (Figure 2). The chloroplast genome organizations were highly conserved across the nine *Cycas* species with only minor variations. Namely, the sizes of IR ranged between 25,003 bp and 25,060 bp across nine *Cycas* species. The rpl23 gene was present only in *C. shiwandashanica* and *C. segmentifida*, and the rpl2 gene was found only in *C. guizhouensis* and *C. crassipes*. With the exception of *C. szechuanensis*, whose trnI gene was positioned far from 43 bp in the LSC, all studied *Cycas* species had the junction LSC/IRb (JLB) located inside the trnI gene. The size and location of the ndhF gene were highly conserved among the nine species, with the same sizes (2219 bp) and the same location at the border of the IRb/SSC junction. The chlL gene at the SSCs was far from the border SSC/IRa (JSA) of both 398 bp and 405 bp. In addition, the psbA gene at the LSCs was far from the border IRa/LSC (JLA) in the range of 152–185 bp.

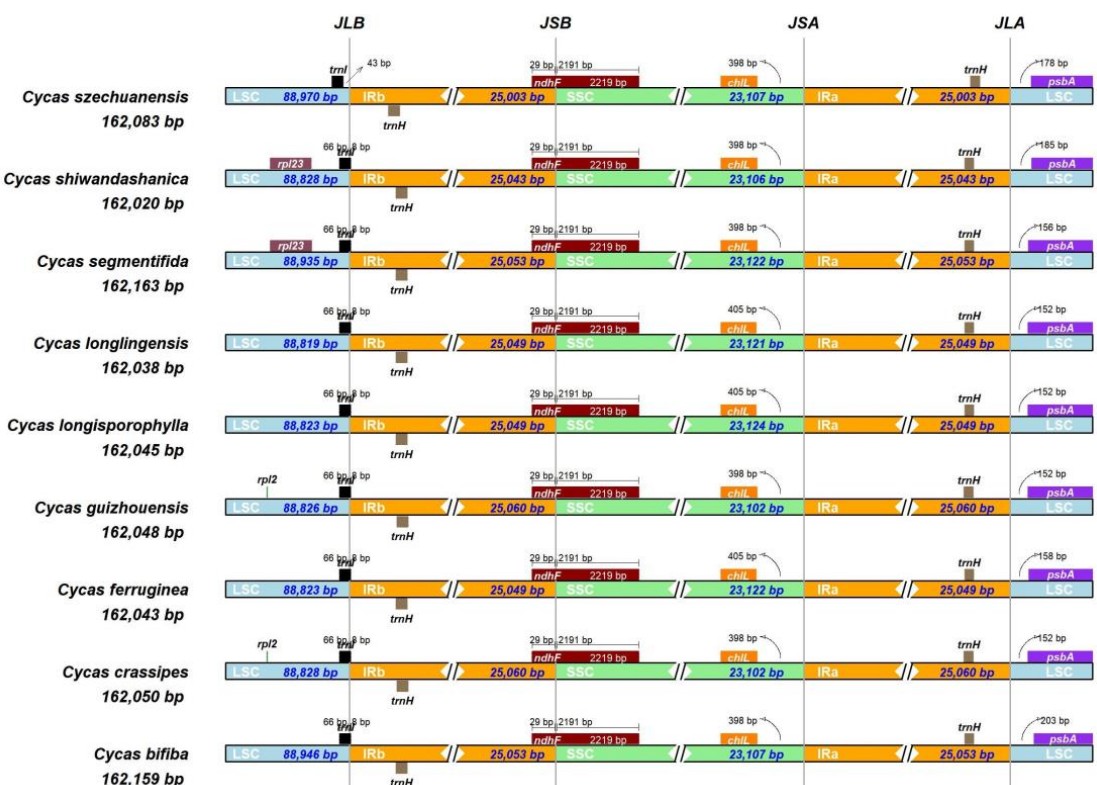

**Figure 2.** Comparison of the IR-SC regional boundaries across chloroplast genomes of nine *Cycas* species. JLB, junction line between LSC and IRb; JSB, junction line between IRb and SSC; JSA, junction line between SSC and IRa; JLA, junction line between IRa and LSC.

### 3.3. Variations and Divergence Hotspot Regions

To assess the extent of sequence polymorphisms, we employed both mVISTA and DnaSP6 tools to compute genetic variations among nine Cycas species. These comparisons involved the entire chloroplast genomes (Figures 3 and 4) in relation to the reference sequence of *C. bifida*. In general, the protein-coding regions in the various species exhibited a high degree of conservation, and highly variable regions were mainly found in intergenic spacers (IGSs) such as psbK-psbI, petN-psbM, trnE-UUC-psbD, ndhC-trnM-CAU, and rpl32-trnP-GGG. These regions of high genetic variation can be utilized for DNA barcode encoding and for conducting phylogenetic analyses within the Cycas genus. The nucleotide variation (Pi) of nine species was negligible, ranging from 0 to 0.0057, with an average value of 0.00104 (Supplementary Table S1). This result agreed with the subtle differences observed in the mVISTA map. The average Pi of the SSC area was 0.00149; that of the LSC area was 0.00126 and that of the IR area was 0.00052.

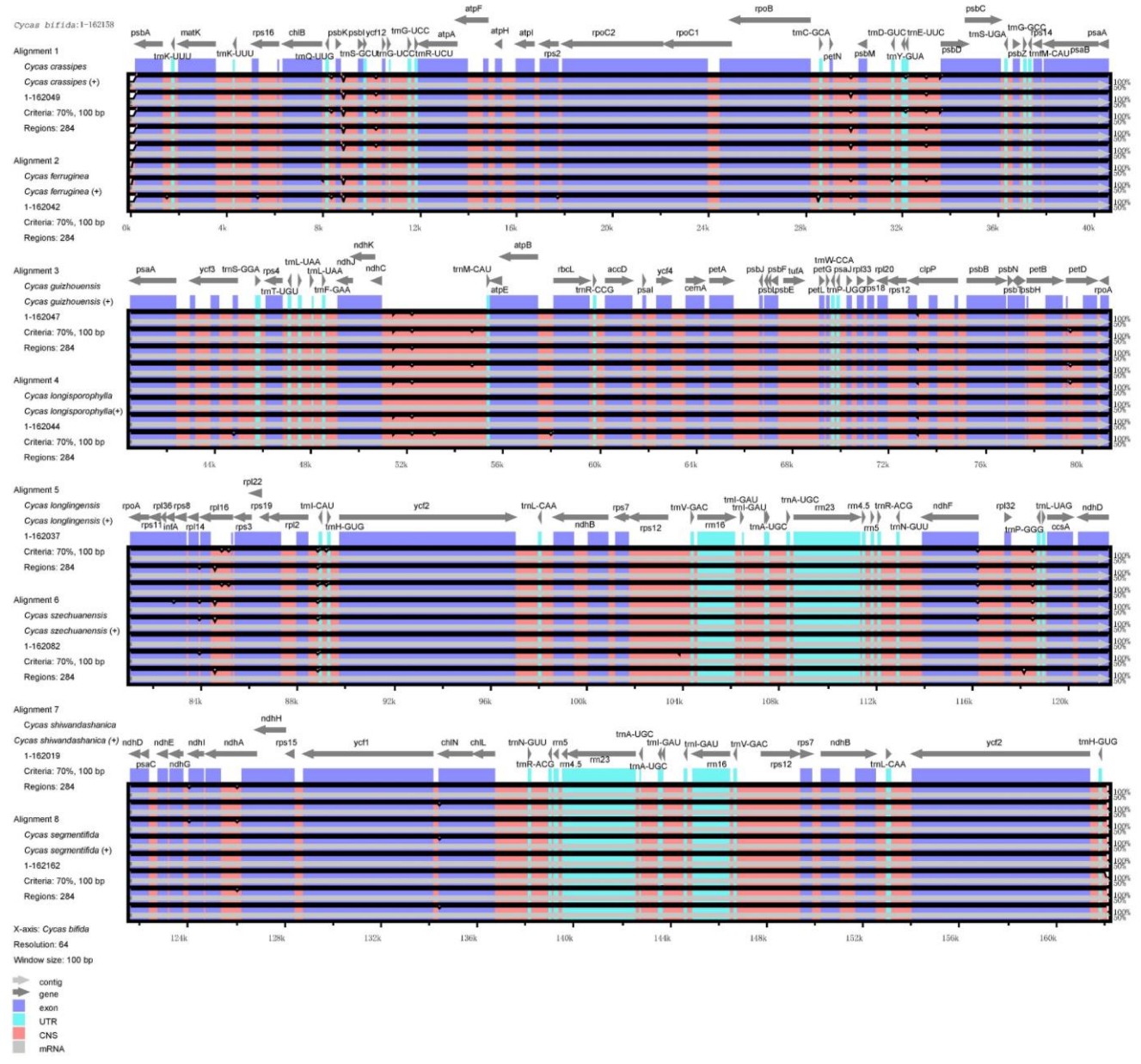

**Figure 3.** Visualization of the alignment of nine chloroplast genomes using mVISTA, with *Cycas bifida* as the reference. The vertical scale represents identity percentages, ranging from 50% to 100%.

Meanwhile, the horizontal scale signifies coordinates within the chloroplast genome. Different regions of the genome are depicted in various colors, distinguishing between exons, introns, and untranslated regions (UTRs) and conserved non-coding sequences (CNSs).

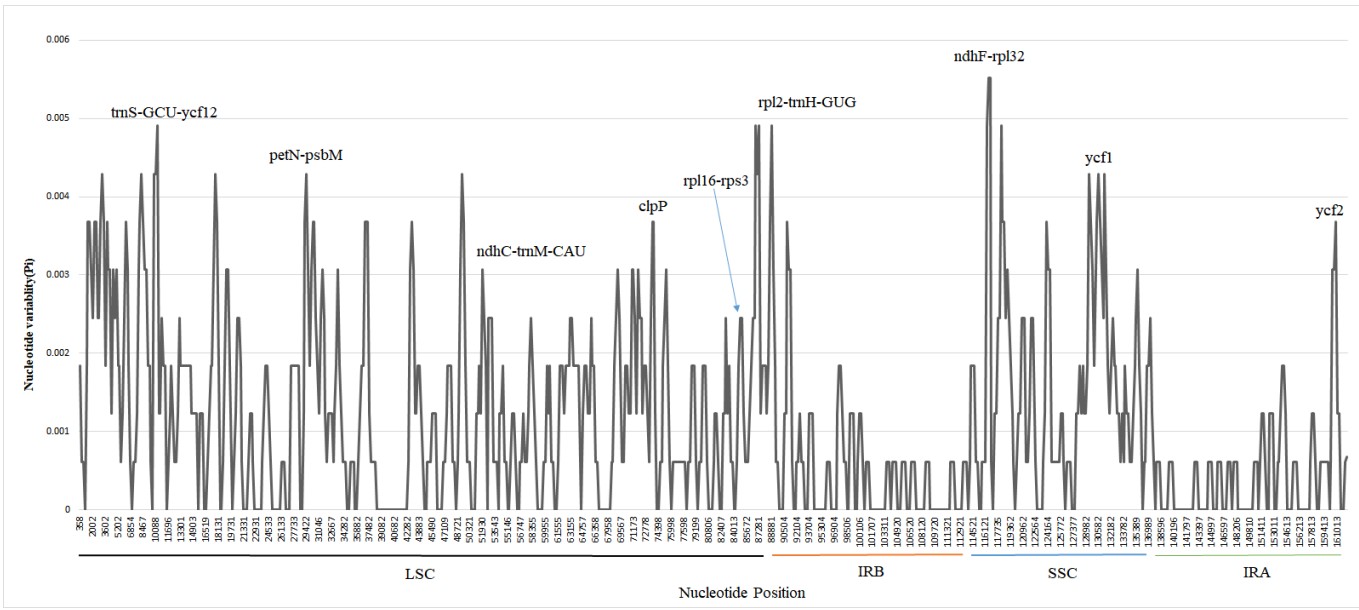

**Figure 4.** Sliding window analysis of nine *Cycas* chloroplast genomes (window length: 600 bp; step size: 200 bp).

### 3.4. SSR and Large Repeats

Simple sequence repeats (SSRs) and large repeats were investigated in nine Cycas cp genomes. The number of SSRs detected was 49–54, and the total quantity of SSR types (mono-/di-/tri-/compound-nucleotide repeats) was detected, with mononucleotide repeats accounting for 77.78%–84.91%. The trinucleotide repeat was only found in *C. guizhouensis* and *C. crassipes* (Figure 5A). At the same time, the distribution of SSRs in the LSC region (76.32%–84.62%) was higher than that in the SSC region (9.09%–15.79%) and IR region (5.00%–10.53%) (Figure 5B). The number of SSRs in the LSC of *C. szechuanensis* and *C. bifida* (37) was the largest, while that of *C. shiwandashanica* and *C. segmentifida* (29) was the lowest.

In this study, an analysis was conducted to identify and quantify all interspersed repetitive sequences within the chloroplast genomes of nine Cycas species, focusing on repeat units with lengths exceeding 20 bp. Additionally, we explored four distinct types of repeats, which encompassed forward repeats (F), inverted repeats (R), complementary repeats (C), and palindromic repeats (P). The analysis revealed that the total repeat number ranged from 47 to 49, with 10–13 forward repeats, 13–17 reverse repeats, 4–6 complementary repeats, and 15–21 palindromic repeats in nine Cycas species (Figure 5C). Interestingly, although the total number of large repeats was similar between species, the length of the repetitive sequence varied distinctly from species to species (Figure 5D). Namely, the repeat length distribution was the same in *C. segmentifida*, *C. longlingensis*, *C. longisporophylla*, and *C. ferruginea*, which contained the most abundant short repeats (21–24 bp) and no long repeats (>33 bp). By contrast, C. g and C. c exhibited no short repeats (21–24 bp) and abundant long repeats (15) greater than 33 bp. In addition, *C. szechuanensis* and *C. bifida* presented similar repeat length distribution patterns, with the highest repeat length being 25–28 bp.

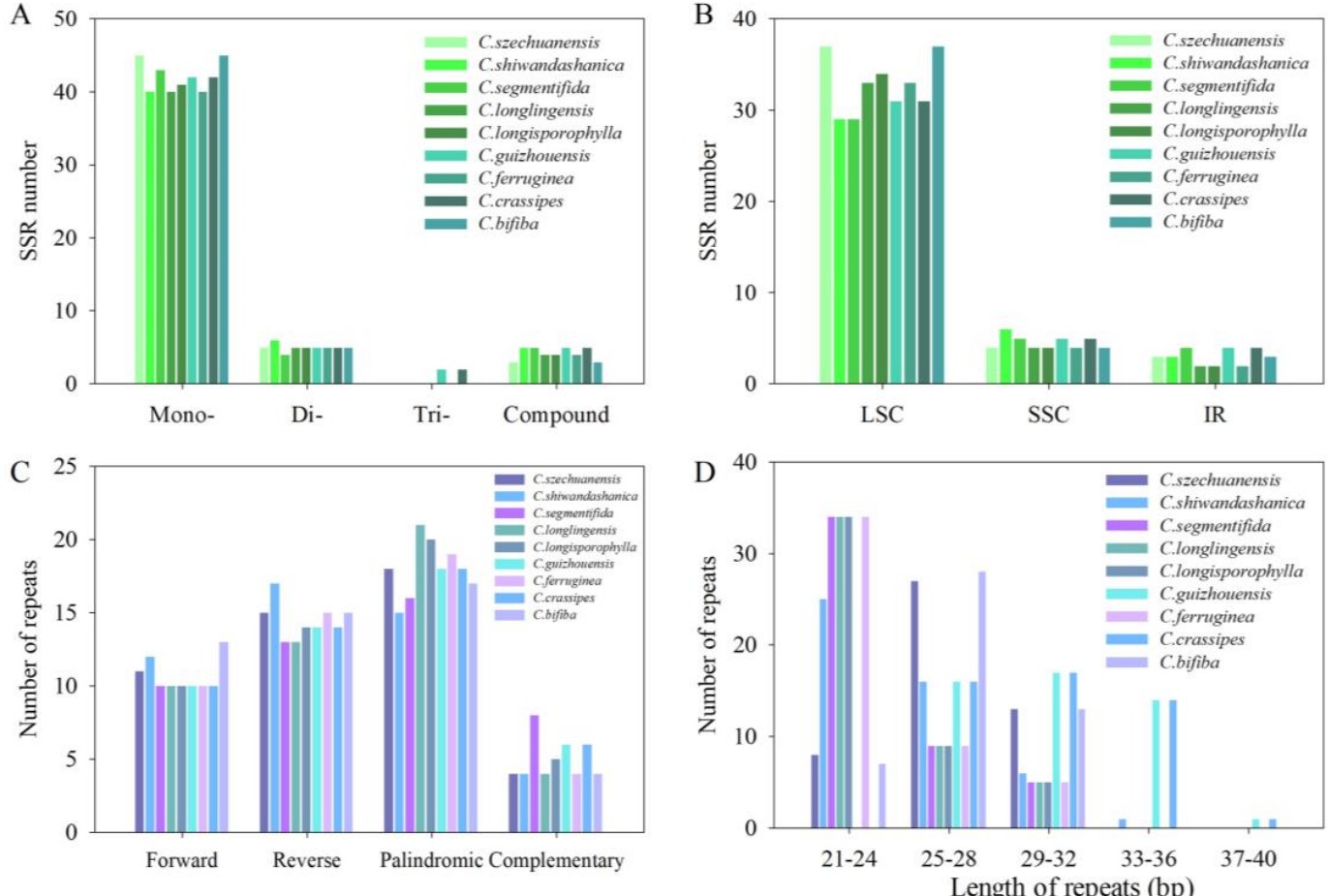

**Figure 5.** Analysis of simple sequence repeats (SSRs) and large repeats in the chloroplast genomes of nine *Cycas* species: (**A**) number of SSRs detected in each species; (**B**) type and frequency of each identified SSR; (**C**) four types of repeats; (**D**) frequency of repeat by length.

*3.5. Phylogenomic Analysis*

In order to gain deeper insights into the phylogenetic positioning of *Cycas* plants and their associations with closely related species, we employed the shared protein-coding genes from the chloroplast genomes of the 14 *Cycas* plants. These genes were utilized to construct a phylogenetic tree via the maximum likelihood (ML) method, which was supported by 1000 bootstrap iterations (Figure 6). The results of the evolutionary tree can be categorized into roughly five distinct sections: *C. taiwaniana*–*C. szechuanensis*, *C. segmentifida*–*C. ferruginea*, *C. debaoensis*–*C. guizhouensis*, *C. panzhihuaensis*–*C. revoluta*, and two outgroup species. Among the six newly sequenced *Cycas* cp genomes, *C. crassipes* and *C. guizhouensis* were sister species. *C. ferruginea* was found to be a sister to *C. longlingensis*, and both were further found to be sister species to *C. longisporophylla*. The cp genes of *C. bifida* presented a close relationship with *C. szechuanensis*.

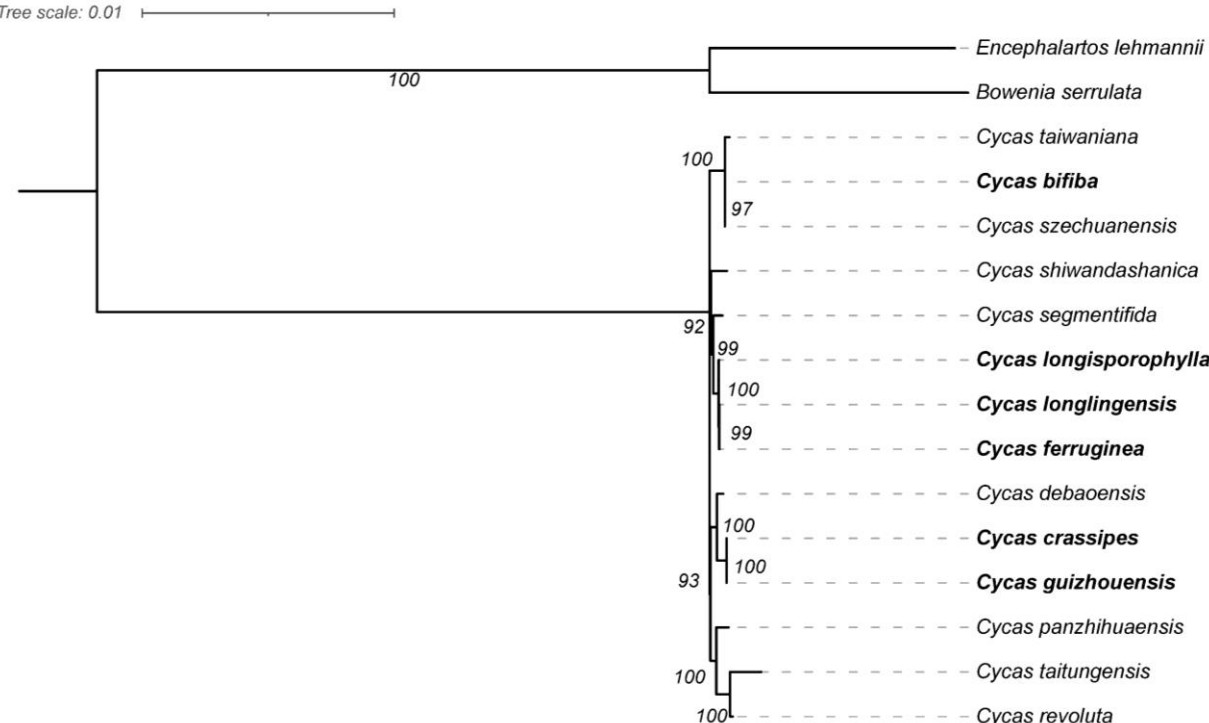

**Figure 6.** Phylogenetic tree of six Cycas species and their related species based on complete chloroplast genomes. The cp genome sequences were downloaded from GenBank.

## 4. Discussion

The chloroplast genome data provide comprehensive insights for examining plant phylogenetics and analyzing evolutionary relationships [21,22]. The abundant data encapsulated within the chloroplast genome render this genome highly suitable as a DNA barcoding tool for species identification [23]. *Cycas*, which is critically threatened across the world [3], has been rarely studied to record its cp genome information. The current work presented the whole cp genomes of six *Cycas* species, four of which (*C. longlingensis*, *C. longisporophylla*, *C. guizhouensis*, and *C. crassipes*) were reported here for the first time. Then, a cp genome comparison was conducted with another three species, *C. szechuanensis*, *C. shiwandashanica*, and *C. segmentifida*, to gain insight into the variations between the aforementioned cp genomes. Together with five other *Cycas* species, a phylogenetic analysis was performed based on complete cp genomes. Studying the cp genome sequences of these species can increase our biological understanding of *Cycas* species' evolution.

In general, plastomes exhibit a high degree of conservation in terms of their genome structures, gene orders, and gene contents [24]. The structural configurations of the entire cp genomes in the six studied *Cycas* species closely resemble those found in the majority of higher plants [25–27]. The overall structure of *Cycas* species is characterized by four distinct regions, including an LSC region spanning from 84,839 to 85,598 bp, an SSC region ranging from 17,559 to 17,687 bp, and two IRs ranging from 31,392 to 31,880 bp each. The comparative examination of six intact cp genomes revealed significant similarities in parameters such as genome length (165,607–167,013 bp), structure, IR/SC borders, and GC content (37.8%–38.0%). In addition, the equal number of rRNA, tRNA, and coding genes indicated that the analyzed *Cycas* species are highly conserved. Previous reports have indicated that GC content exhibits variation across distinct regions of cp genomes, with the IR regions displaying higher GC content due to the inclusion of rRNAs [25,28], which is in line with our results.

By comparing the variations in cp genome sequences between distinct taxa, it becomes possible not only to efficiently identify DNA fragments rich in information but also to foster the advancement of techniques for species identification and the exploration of population

diversity [29]. mVISTA and DnaSP6 were applied to evaluate variations in the cp genomes of different *Cycas* species, and both methods demonstrated that *Cycas* cp genomes were highly conserved. The IR regions were less variable than the LSC and SSC regions, which was consistent with the findings of a prior investigation [30]. In addition, a prior report indicated higher susceptibility to mutations in non-coding regions compared to coding regions [31]. In the present study, we observed only high variable regions within IGSs, rather than coding regions, which aligns with this characteristic.

Inheritance of the cp genome is uniparental, and, within a given species, there exists a notable degree of variation in SSRs [32]. Consequently, these SSRs serve as valuable molecular indicators for developmental analyses and species identification purposes [33]. Moreover, SSRs frequently find applications as genetic markers in investigations pertaining to community genetics and evolutionary research [34]. Among the repeats, those that showed the greatest enrichment were mononucleotide repeats followed by dinucleotide repeats. Overall, trinucleotide repeats were infrequent across all nine *Cycas* cp genomes. When conducting a comparative assessment of repeat sequences within the cp genomes, the repeat length distribution was the same in *C. segmentifida*, *C. longlingensis*, *C. longisporophylla*, and *C. ferruginea*, with an average length of 24.146 bp. In contrast, *C. guizhouensis* and *C. crassipes* exhibited much longer repeats, with an average of 30.563 bp. Notably, species with similar repeat length distributions of their cp genomes were close in the phylogenetic tree, indicating that large repeats are reliable molecular indicators in evolutionary studies.

Currently, protein-coding genes are commonly used to build phylogenetic trees [35]. The results of this study revealed the genetic relationships between *Cycas* plants. According to Zheng et al., *C. revoluta* and *C. taitungensis* belong to Asiorientales, while *C. panzhihuaensis* belongs to Panzhihuaenses [6]. Consistent with this classification, the first two species were also grouped into the same clade in our results and further into a clade with *C. panzhihuaensis*. This relationship was also consistent with the research results of Yang et al. [36]. Additionally, Zheng et al. discovered 18 species of Stangerioides in China [6], but the *C. bifida*, *C. longisporophylla*, and *C. longlingensis* species analyzed in this study were not included in their research. Here, the phylogenetic tree indicates that these three species should belong to Stangerioides since they are grouped together in the same branch as other species that are part of this section. This categorization is justified for two reasons. Morphologically, the testa coats of this species are yellow to brown, and their microsporangiate cones and microsporophylls are soft to the touch. Geographically, these species are all found in the Guangxi area of China [37,38]. In summary, our phylogenetic analysis of *Cycas* species relied upon protein-coding genes, which currently constitute the most comprehensive dataset available. This endeavor not only lays the theoretical groundwork in this field but also provides the requisite technical details for advancing and effectively utilizing resources derived from *Cycas* plants.

## 5. Conclusions

In summary, our investigation into the evolutionary relationships within the *Cycas* genus was enhanced by the assembly and annotation of chloroplast genomes from six distinct *Cycas* species, coupled with comparative analyses. The *Cycas* cp genomes were characterized by four distinct regions and span from 162,038 to 162,159 bp. They were highly conserved, and several non-coding sites (psbK-psbI, petN-psbM, trnE-UUC-psbD, ndhC-trnM-CAU, and rpl32-trnP-GGG) showed significant variation. These findings have contributed to our understanding of the genetic diversity and evolutionary processes within this plant family, as reflected in aspects like genome structure, GC content, nucleotide diversity, and sequence divergence. The chloroplast genomes of the six *Cycas* species offer valuable resources for the development of genomic markers, which, in turn, can bolster future genetic research endeavors, support conservation initiatives, and facilitate phylogenetic analyses within the Cycadaceae family.

**Supplementary Materials:** The following supporting information can be downloaded at: https://www.mdpi.com/article/10.3390/f14102069/s1; Table S1: The nucleotide variation (Pi) of nine *Cycas* species.

**Author Contributions:** Conceptualization, J.T.; methodology, R.Z.; validation, T.C.; formal analysis, S.Z. and L.P.; investigation, T.D. (sampling), S.C. (sequencing), R.Z. (data analysis); writing—original draft preparation, J.T.; writing—review and editing, T.C.; visualization, L.P. and S.Z.; supervision, X.W.; funding acquisition, X.W. All authors have read and agreed to the published version of the manuscript.

**Funding:** This study was supported by the National Key Research and Development Program (No.2022YFF1300700), Guangxi Natural Science Foundation of China (No. 2020GXNSFAA259029), Chinese Academy of Sciences 'Light of West China' Program (2022), Guangxi Forestry Science and Technology Promotion Demonstration Project (2023LYKJ03 and [2022]GT23), Guangxi Key Laboratory of Plant Functional Phytochemicals Research and Sustainable Utilization Independent Project (No. ZRJJ2022-2), Guilin Innovation Platform and Talent Plan (20210102-3), and Guilin City Technology Application and Promotion Plan (20220134-3).

**Data Availability Statement:** All data cited in the study are publicly available.

**Conflicts of Interest:** The authors declare no conflict of interest.

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
