# Peer review of "Comparative Analysis of the Complete Chloroplast Genomes of Six Endangered Cycas Species: Genomic Features, Comparative Analysis, and Phylogenetic Implications"

_forests, doi:10.3390/f14102069_

Round 1

Reviewer 1 Report

·       Avoid unnecessary information in the abstract of the article.

·       Write objectives should be clear and concise.

·       Repeat the 51 and 54 sentences

·       Results of the research work are well, but clearly explained in the result and discussion section.

·       Discussion must be connected with results parts.

·       Remove the seed plants, and write gymnosperm plants (160-161 sentences).

·       Please follow this article and cite it (Comparative genome sequence and phylogenetic analysis of chloroplast for evolutionary relationship among Pinus species). Please, help with the result part.

·       Increase the fig 1 resolution. Not show cleared.

·       The phylogenetic tree of six Cycas species and their related species based on complete chloroplast genomes, the tree does not show clearly. Please improve and increase the resolution of the tree. The tree must be compared to previous studies

·       Write the conclusion section separately.

Author Response

1. Avoid unnecessary information in the abstract of the article.

Response: we have revised the abstract and avoided unnecessary information.

2. Write objectives should be clear and concise.

Response: we have clearly stated the objectives in the revised version.

3. Repeat the 51 and 54 sentences

Response: sorry that we are confused with this comment. Do you mean line 51 and line 54 are repeated? In line 51, we wanted to say that it is challenging in Cycas categorization. While in line 54, we showed the reason for challenging.

4. Results of the research work are well, but clearly explained in the result and discussion section. Discussion must be connected with results parts.

Response: we have connected the discussion part with result part and highlighted the connection sentences.

5. Remove the seed plants, and write gymnosperm plants (160-161 sentences).

Response: thanks for the suggestion. We have revised accordingly.

6. Please follow this article and cite it (Comparative genome sequence and phylogenetic analysis of chloroplast for evolutionary relationship among Pinus species). Please, help with the result part.

Response: yes, we have cited the reference as you suggested.

7. Increase the fig 1 resolution. Not show cleared.

Response: We have re-uploaded figure 1.

8. The phylogenetic tree of six Cycas species and their related species based on complete chloroplast genomes, the tree does not show clearly. Please improve and increase the resolution of the tree. The tree must be compared to previous studies

Response: We have re-uploaded figure 1 and compared the tree with previous studies in the discussion section.

9. Write the conclusion section separately.

Response: we have added the conclusion section in the revised manuscript.

Reviewer 2 Report

In this study, the cp genomes of six Cycas species were sequenced with the potential to provide a detailed analysis of its assembly and annotation. The authors, through a comparative analysis, highlight how the Cycas chloroplast genome was highly conserved, but there were some non-coding sites with significant variations.

The article is well written, however the figures presented in the main body are absolutely illegible and this determines a lack of support for the results.

Below are some considerations. 

1)                 Line 46-48. “There is only one genus of Cycadaceae, Cycas, which contains roughly 120 species [4]. This genus contains a large number of species, involves a complicated taxonomy, and covers a wide geographical range”. The sentence in line 46 and the beginning of that in 47 could be integrated into a single sentence since they express the same concept. Review its construction.

2)                  Line 85: provide a space between the parts of the text, as required by the magazine format.

3)                 Figure 1-The format included in the main text is unfortunately difficult to read as it is out of focus. No additional figures of better quality were provided. Please provide readable format figures.

4)                 Figure 2- with the exception of species and JLB, JSB, JSA, JLA at the top, the rest of the figure is completely illegible. Please provide a figure that is functional for reading the results, otherwise do not include it. So, all the results reported from line 183 to 189 cannot be seen in the figure.

5)                 Figure 3 is absolutely unusable. Please provide better quality to find a way to replace it with a readable form.

6)                 This also applies to figure 4 and figure 5: for which an effort of free interpretation must be made. Please provide adequate material.

Best Regards.

Author Response

1) Line 46-48. “There is only one genus of Cycadaceae, Cycas, which contains roughly 120 species [4]. This genus contains a large number of species, involves a complicated taxonomy, and covers a wide geographical range”. The sentence in line 46 and the beginning of that in 47 could be integrated into a single sentence since they express the same concept. Review its construction.

Response: we have rewritten the sentence in the revised version.

2) Line 85: provide a space between the parts of the text, as required by the magazine format.

Response: we have reformatted the manuscript.

3) Figure 1-The format included in the main text is unfortunately difficult to read as it is out of focus. No additional figures of better quality were provided. Please provide readable format figures.

Response: We have thoroughly examined the issue you raised regarding the image resolution. It appears that the pixel quality of the images in the Word version is indeed adequate. The reduction in image quality likely occurred during the PDF conversion process. We have rectified this by re-uploading the document and images into the system, ensuring that the images are now presented in a clear and high-resolution format.

4)Figure 2- with the exception of species and JLB, JSB, JSA, JLA at the top, the rest of the figure is completely illegible. Please provide a figure that is functional for reading the results, otherwise do not include it. So, all the results reported from line 183 to 189 cannot be seen in the figure.

Response: We have re-uploaded figure 2.

5) Figure 3 is absolutely unusable. Please provide better quality to find a way to replace it with a readable form.

Response: We have re-uploaded figure 3.

6)  This also applies to figure 4 and figure 5: for which an effort of free interpretation must be made. Please provide adequate material.

Response: We have re-uploaded figure 4 and 5.